# On the Powerfulness of Textual Outlier Exposure for Visual OoD Detection

**Sangha Park**[1], **Jisoo Mok**[1], **Dahuin Jung**[1], **Saehyung Lee**[1], **Sungroh Yoon**[1,2,*]

[1]Department of Electrical and Computer Engineering, Seoul National University
[2]Interdisciplinary Program in Artificial Intelligence, Seoul National University
`{wiarae, magicshop1118, annajung0625, halo8218, sryoon}@snu.ac.kr`

## Abstract

Successful detection of Out-of-Distribution (OoD) data is becoming increasingly important to ensure safe deployment of neural networks. One of the main challenges in OoD detection is that neural networks output overconfident predictions on OoD data, make it difficult to determine OoD-ness of data solely based on their predictions. Outlier exposure addresses this issue by introducing an additional loss that encourages low-confidence predictions on OoD data during training. While outlier exposure has shown promising potential in improving OoD detection performance, all previous studies on outlier exposure have been limited to utilizing visual outliers. Drawing inspiration from the recent advancements in vision-language pre-training, this paper venture out to the uncharted territory of textual outlier exposure. First, we uncover the benefits of using textual outliers by replacing real or virtual outliers in the image-domain with textual equivalents. Then, we propose various ways of generating preferable textual outliers. Our extensive experiments demonstrate that generated textual outliers achieve competitive performance on large-scale OoD and hard OoD benchmarks. Furthermore, we conduct empirical analyses of textual outliers to provide primary criteria for designing advantageous textual outliers: near-distribution, descriptiveness, and inclusion of visual semantics. Code is available at `https://github.com/wiarae/TOE`

## 1 Introduction

The standard assumption when deploying neural networks is that both the train and test data will fall under the same distribution, in which case, test data are considered to be in-distribution (ID). However, in the real world, they often encounter out-of-distribution (OoD) data, which refers to data that are distant from the training data distribution. Because the generalization capacity of neural networks does not extend outside the distribution of observed train data, OoD data can cause them to perform poorly or even fail [11]. Therefore, it is important for neural networks to not only achieve high accuracy on ID data but also successfully detect OoD data to preclude them from the inference process. Unfortunately, neural networks yield highly over-confident predictions on OoD data, making OoD detection a non-trivial research problem [39, 43].

One promising approach to OoD detection is "outlier exposure," which modifies the training procedure of neural networks [23, 18], such that they can accurately classify ID data while also reliably detecting OoD data. To simultaneously accomplish the two training objectives, outlier exposure utilizes proxy data that can emulate actual OoD data that the neural network may confront at test time. Outlier exposure then defines a regularization term that encourages low-confidence predictions on those proxy data. Finally, the neural network is optimized to minimize the regularization term in addition

---

*Corresponding author

to the supervised loss. The major challenge in outlier exposure is providing proxy data for training without explicit knowledge of OoD data.

Previously when neural networks could only process single-modal data, outlier exposure could only rely on visual outliers from the image domain. The recent emergence of multi-modal neural networks [37, 21] has opened new research opportunities. In this work, we study the powerfulness of textual outlier exposure by utilizing multi-modal neural networks. We first conduct preliminary studies of textual outlier exposure by replacing visual outliers with their textual counterparts in two widely-studied outlier exposure methods: real and virtual outliers. Our empirical results reveal that visual outliers suffer from high performance variation according to the choice of an auxiliary dataset and non-negligible time consumption. Textual outliers, on the contrary, do not experience the above problems, demonstrating that they can serve as attractive proxy data for outlier exposure.

Textual outliers can be designed in various forms, from single words to detailed descriptions. Initially, it may be unclear which type of textual outlier is most effective for facilitating textual outlier exposure. To address this ambiguity, we explore textual outliers at three verbosity levels: word, description, and caption. These textual outliers are generated using large-language models (LLMs) such as GPT-3, BERT, or vision-language models (VLMs) like BLIP-2. Among generated outputs, only those that can be considered OoD are utilized for outlier exposure. Our textual outlier generation methods rely solely on images and class labels of ID data. As a result, utilizing generated textual outliers eliminates the need to collect and curate proxy data to perform outlier exposure.

Our comprehensive experimental results present compelling evidence that the proposed textual outlier approach outperforms existing methods for OoD detection. Specifically, compared to the method in Hendrycks et al. [18] that employs real auxiliary datasets of outliers, our caption-level textual outlier reduces the FPR95 from 73.80% to 58.21%, yielding a direct improvement of 15.59%p on the challenging ImageNet-1K benchmark [6]. Additionally, our textual outlier approach surpasses advanced baselines in visual outlier exposure, demonstrating the advantages of the outlier synthesis approach in the textual space. Furthermore, through comparative analyses of visual versus textual outliers and different forms of textual outliers, we establish three key criteria for advantageous textual outliers: near-distribution, descriptiveness, and inclusion of visual semantics.

Our contributions are summarized as follows:

- This is the first work to investigate the potential of textual outlier exposure with multi-modal neural networks. We reveal that textual outlier exposure can outperform its visual counterparts and overcome inherent limitations associated with them.

- We propose to utilize LLMs or VLMs to generate diverse drafts of textual outliers at different verbosity levels: word, description, and caption. Generative models facilitate the convenient and cost-effective collection of potential texts suitable for formatting into textual outliers. The generated outputs are then refined into more effective forms for textual outlier exposure.

- Proposed textual outliers are validated in various OoD detection settings through comparison with visual outlier exposure baselines. In particular, description-level textual outlier outperforms a competitive baseline from visual outlier exposure by 38.60%p (AUROC) on a challenging detection scenario with semantically similar classes.

## 2 Problem Setup

**Background and Notations.** Let $\mathcal{X}$ represent the input space, and $\mathcal{Y} = \{1, ..., C\}$ denote the label space. In our framework, we consider two main datasets: the ID dataset, $D_{\text{ID}}$, and the OoD dataset, $D_{\text{OoD}}$. The $D_{\text{ID}} = \{(\mathbf{x}_i, y_i)\}_{i=1}^n$ is a set of every sample from ID and is defined over $\mathcal{X} \times \mathcal{Y}$. On the other hand, $D_{\text{OoD}}$ is defined solely over the input space $\mathcal{X}$ and does not have any associated labels. During training, the neural network never observes the OoD dataset. In the inference process, the neural network should be able to reliably filter out OoD samples.

**Out-of-distribution detection** can be formulated as a binary classification problem. Given a classifier $f : \mathcal{X} \to \mathbb{R}^C$ trained on samples in a subset of $D_{\text{ID}}$, the objective is to design a binary function estimator,

$$g(\mathbf{x}) = \begin{cases} \text{in,} & \text{if } S(\mathbf{x}, f) \geq \gamma \\ \text{out,} & \text{if } S(\mathbf{x}, f) < \gamma \end{cases} \tag{1}$$

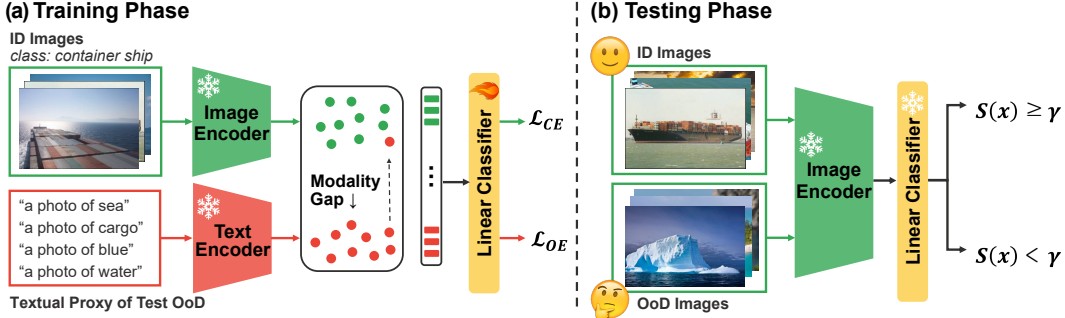

Figure 1: Framework overview. (a) During the training phase, we utilize textual outliers embeddings as proxies for test OoD data. For the ID image data, we train the model using cross-entropy loss, while for the textual outlier data, we employ outlier exposure loss, which enforce the model to output the low-confidence predictions on them. To reduce the modality gap between the two domains, we inject noise to textual outlier. (b) During the testing phase, the trained classifier filters out OoO data by thresholding output score.

that classifies whether a sample $\mathbf{x} \in \mathcal{X}$ belongs to $D_{\text{ID}}$ or not. This is determined by the score function $S(\mathbf{x}, f)$, which is defined based on the classifier's output. The threshold parameter $\gamma$ is typically chosen to ensure a high percentage (e.g., 95%) of correctly classified in-distribution samples.

**Evaluation Metrics.** The OoD detection performance can be assessed by using the following two evaluation metrics: (1) the false positive rate of OoD samples at a fixed true positive rate of 95% (FPR95) for ID samples, with lower values indicating better performance, and (2) the area under the receiver operating characteristic curve (AUROC), with higher values indicating better performance. Additionally, we report the accuracy of the ID classification.

**Framework Overview.** CLIP [37] is a large-scale vision-language model that has been trained on a vast dataset consisting of 400 million image-text pairs, collected from the internet. Its remarkable performance has been demonstrated across various tasks, leading to numerous studies exploring its transferable feature representation [59, 13]. Several previous works have shown improvements in diverse tasks by operating in the CLIP space with the frozen CLIP encoder [42, 14, 15, 9, 35, 1]. Likewise, we leverage the strength of the CLIP encoder to build a reliable classifier by training a linear classifier on top of it. The CLIP embedding space that jointly models images and texts enables usage of textual cues for visual OoD detection. Therefore, we employ textual, instead of visual, outliers to regularize the classifier; train data for the supervised loss and test data, both ID and OoD, remain defined in the image domain. The general framework for textual outlier exposure is illustrated in Figure 1.

## 3 Understanding the Potential of Textual Outlier

Prior to exploring various forms of textual outliers, we first show that even naïve approaches to textual outlier exposure can overcome the inherent limitations of its visual counterpart. We reveal the advantages of using textual outliers over images by replacing images with texts in two representative types of outlier exposure: (1) the use of real auxiliary datasets and (2) virtual outlier synthesis in the feature space. The empirical results demonstrate that textual outliers can achieve superior OoD detection performance, as well as significantly reducing the variance of performance and time consumption in (1) and (2). In this section, all of the experiments are conducted following the procedure in Figure 1.

### 3.1 Auxiliary datasets

**Experimental setting.** Hendrycks *et al.* [18] proposed to utilize real auxiliary datasets as proxy data for outlier exposure to improve the detection performance. We leverage four auxiliary datasets to evaluate visual and textual outliers (Table 1). To use texts instead of images, we adopt class labels from an auxiliary dataset as outliers, *e.g.,* "a photo of *bamboo forest*" when SUN [52] is used as an

Table 2: Results of using images *vs.* texts (class labels) of auxiliary datasets as outliers. We compare the results of utilizing 4 different auxiliary datasets. In the testing phase, we use ImageNet20 and ImageNet10 as ID and OoD datasets, respectively. The best result in each column is in bold.

| Auxiliary Dataset | Texture | | Places | | SUN | | iNaturalist | | | Average | |
|---|---|---|---|---|---|---|---|---|---|---|---|
| Metrics | FPR | AUROC | FPR | AUROC | FPR | AUROC | FPR | AUROC | ID Acc | FPR | AUROC |
| None | - | - | - | - | - | - | - | - | 95.2 | 21.49 | 96.27 |
| Image | 29.80 | 95.96 | **13.60** | **97.53** | 18.60 | 96.89 | 36.20 | 93.34 | 95.6 | $24.55_{\pm 10.05}$ | $95.93_{\pm 1.84}$ |
| **Text** | **19.60** | **97.23** | 15.40 | 97.37 | **15.00** | **97.37** | **13.40** | **97.28** | **95.9** | $\mathbf{16.10}_{\pm 3.12}$ | $\mathbf{97.25}_{\pm 0.17}$ |

auxiliary dataset. To mitigate the imbalance in size between image and text datasets, we, following the experimental setting in Zhang *et al.* [57], train the model for 300 epochs with textual outliers and 25 epochs with visual outliers. We use ImageNet20 as ID data and ImageNet10 as OoD data for evaluation as proposed in Ming *et al.* [32].

**Results.** Table 2 presents a comparison of the detection performance and classification accuracy between visual and textual outliers. Notably, textual outliers demonstrate superior performance across all evaluation metrics. Furthermore, the detection performance of models trained with visual outliers is highly contingent on the choice of an auxiliary dataset, leading to significant performance variation. This poses a considerable challenge in determining the optimal dataset for effective outlier exposure. In contrast, the use of textual outliers noticeably reduces performance variation. Additional results on other benchmark datasets can be found in the Appendix C.1.

Table 1: Number of images / texts (class labels) in auxiliary datasets.

| | Textures | Places | SUN | iNaturalist |
|---|---|---|---|---|
| Image | 5640 | 10000 | 10000 | 10000 |
| Text | 47 | 50 | 50 | 110 |

## 3.2 Synthesis in feature space

**Experimental setting.** As an alternative to using auxiliary datasets, virtual outlier synthesis in the feature space, which does not rely on real datasets, has been proposed [8, 46]. The feature space of a classifier can be represented as a Gaussian mixture model with the class mean and variance. It is reasonable to consider instances sampled from the low-likelihood regions of this distribution as outliers. To obtain means and variances per class using text instead of images, we adopt class labels, synonyms of class labels, and descriptions of classes obtained from GPT-3 [2]. Examples of labels and associated synonyms and descriptions are provided in the Appendix C.2. We choose VOS [8] from a family of methods in outlier synthesis as the testbed to compare virtual outliers. For both textual and visual outliers, we utilize the CLIP embedding space to obtain class means and variances. We train the linear classifier with ImageNet10 and use ImageNet20 as OoD dataset for evaluation [32]. The number of training epochs is set to 300 for both images and texts.

Table 3: AUROC achieved after varying amounts of time consumption for virtual outlier synthesis.

| | Time | | |
|---|---|---|---|
| | 18m | 30m | 3h 54m |
| Image | 91.65 | 96.59 | 97.77 |
| **Text** | **97.85** | - | - |

**Results.** Our findings, as shown in Table 3, indicate that utilizing class means and variances derived from texts for outlier synthesis outperforms those derived from images, while also demonstrating better sample efficiency. The majority of computational overhead in virtual outlier synthesis arises from computing the updated class mean and variance after each epoch to generate a new set of virtual outliers. Notably, when using text data, the training time is significantly reduced by a factor of 13 (18 minutes *vs.* approximately 4 hours), while still achieving comparable performance. When we reduce the amount of image data to match the synthesis and training time for textual outliers, the OoD detection performance experiences a considerable decline from 97.77 to 91.65 in terms of AUROC.

## 3.3 Analysis

The superior performance of textual outliers can be attributed to their proximity to ID data. To analyze the relationship between ID data and different outliers, we utilize UMAP [30], a popular technique for dimension reduction. In Figure 2 (a), we visualize the latent space that con-

tains ID data and two types of real auxiliary datasets, Texture [4] and iNaturalist [47]. They are the two auxiliary datasets whose textual outliers led to greatest performance improvements. Similarly, Figure 2 (b) shows the latent space representing three types of data: ID data, class-wise means of ID data in both image and text domains, and outliers sampled from each domain's distribution. Textual outliers, whether real or virtual, lie closer to ID data compared to their visual counterparts. This observation implies that textual outliers, which reside in the vicinity of ID data distribution, can offer informative signals about boundary regions [8, 33].

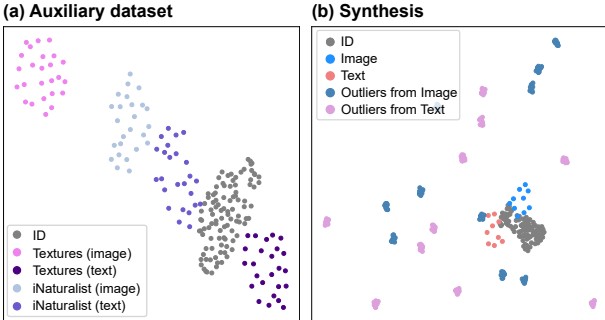

As a result, the usage of textual outliers contributes to improving OoD detection performance by assisting the neural network in modeling the ID-OoD boundary.

Based on these empirical findings, we conclude that exposing the neural network to textual outliers offers significant advantages, including robustness to performance variation and enhanced sample efficiency. In the subsequent sections, we delve into potential designs for textual outliers (Section 4) and analyze the factors that contribute to their effectiveness in outlier exposure (Section 5).

Figure 2: Feature space analysis of image and text outliers in (a) auxiliary datasets and (b) synthesis settings.

## 4 Method

Unlike visual outliers, textual outliers can take various forms, ranging from a single word to more verbose descriptions. We investigate three different designs for textual outliers: 1) words retrieved from ID images, 2) descriptions of ID class-labels, and 3) captions of ID images generated by an image captioning model. Figure 3 visualizes the generation process for each type of textual outlier. We utilize LLMs and a VLM to generate textual outliers and selectively employ the generated outputs that are OoD but still lie adjacent to the ID data. We highlight that no additional data except for ID data is used to generate textual outliers.

### 4.1 Generating textual outliers in three types

#### 4.1.1 Word-level textual outliers

**(i) Motivation.** The most straightforward way of designing textual outliers is in the form of `"a photo of {word},"` which we coin word-level textual outliers. As discussed in Section 3.1, word-level textual outliers can generated by simply leveraging class labels or comparable terms from auxiliary datasets, but they may not always lie in the proximity of ID data. In Section 3.3, we have observed that textual outliers that are closer to the ID data tend to provide more informative signals. To obtain word-level textual outliers that are near the ID data, we utilize image-based text retrieval. Utilizing texts that are close to images enhances the classifier's ability to capture the ID-OoD boundary, thereby improving its performance in distinguishing between ID and OoD data.

**(ii) Generation.** We exploit the framework proposed by Esmaeilpour *et al.* [10] to perform text retrieval from ID images. The retrieval process consists of the following steps. First, we use the CLIP image encoder to extract image embeddings of ID data. These embeddings are then treated as sequences of tokens and fed into the multi-head cross-attention head of BERT, which functions as a text decoder. Finally, the resulting text embeddings are linearly projected onto the text space. Figure 3 (a) provides a high-level overview of the retrieval process. Images from the validation set of the ID dataset, which does not overlap with the test set, are used as inputs.

Table 4: Templates for prompt ensembling with textual outliers.

| |
|---|
| `"A photo of a {}"` |
| `"A blurry photo of a {}"` |
| `"A photo of many {}"` |
| `"A photo of the large {}"` |
| `"A photo of the small {}"` |

**(iii) Filtering.** Initial versions of generated text outputs are not always guaranteed to be OoD. Therefore, we select a subset of retrieved words based on their cosine similarity to the ID image

**(a) Word-level Textual Outlier**

Images from ID Validation Set → [CLIP] Image Encoder → [BERT] Text Decoder → Linear Projection → **Top-k{** Fur Animal Tiger Nose Feline Mammal**} ∩ {Class Labels}$^c$**

**(b) Description-level Textual Outlier**

What are useful features for distinguishing a [siamese cat] in a photo?
Category Names from ID Validation Set

→ [GPT-3] → blue eyes long, slender body short fur

**(c) Caption-level Textual Outlier**

Step 1. Caption generation

Images from ID Validation Set → [BLIP-2] → "a Siamese cat with blue eyes sitting on the floor" / "two cats lying on a couch."

Step 2. Outlier caption selection

Inlier: "a Siamese cat with blue eyes sitting on the floor"

Siamese Cat

"two cats lying on a couch." Outlier

Figure 3: Illustration of (a) word-level, (b) description-level, and (c) caption-level textual outlier generation processes. A different generative model is used for each type of textual outlier. In (a), CLIP and BERT are used to perform image-based text retrieval to create word-level textual outliers. In (b) and (c), description-level and caption-level textual outliers are generated from GPT-3 and BLIP-2, respectively. For word- and caption-level textual outliers, the generated outliers are refined for the purpose of outlier exposure. See Section 4 for details.

embeddings. Only those with top-$k$ to $k + \delta$ similarity values are selected, and afterwards, class labels are explicitly removed. Through this filtering process, we can effectively keep texts that are OoD but near ID data. To prompt CLIP with selected word-level textual outliers, we employ prompt ensemble using the templates outlined in Table 4.

### 4.1.2 Description-level textual outliers

**(i) Motivation.** We now consider generating textual outliers from class labels of ID data, instead of images. With LLMs, class labels can easily be expanded into illustrative descriptions that include linguistic semantics; we name these description-level textual outliers.

**(ii) Generation.** We utilize GPT-3, a powerful and widely-adopted LLM, to obtain class descriptions. GPT-3 is prompted with the phrase "`describe what the {class name} looks like`" [31].

**(iii) Filtering.** These class descriptions can be utilized as textual outliers without further refinement because they already are OoD from CLIP's viewpoint. For instance, when we prompt CLIP with a description of a Siamese cat (*e.g.,* "`a photo of` + *light-colored fur with dark points on the face, ears, legs, and tail*"), it retrieves images that noticeably deviate from the actual visual characteristics of a Siamese cat. An illustrative example demonstrating this phenomenon can be found in the Appendix E.1. Therefore, we employ the class descriptions obtained from GPT-3, after excluding the corresponding class labels, as textual outliers and prepend templates in Table 4 to each description.

### 4.1.3 Caption-level textual outliers

**(i) Motivation.** To generate descriptive textual outliers that also contain visual semantics, we propose to reformulate captions into textual outliers. The so-called caption-level textual outliers, generated with visual cues, can capture the complex visual semantics in images and also contain rich linguistic semantics, encompassing the benefits of word- and description-level textual outliers.

**(ii) Generation.** To acquire a set of captions, denoted as $\mathcal{S} = \{\mathbf{s}_1, \mathbf{s}_2, ..., \mathbf{s}_N\}$, we utilize BLIP-2 [25], known for its exceptional descriptive capabilities [40]. Images of the validation set of the ID data are used as inputs to BLIP-2. However, some of the captions that resemble the ID samples too closely are ineffective for outlier exposure.

**(iii) Filtering.** To filter out such ineffective captions, we utilize the Mahalanobis distance $\mathrm{M}(\mathbf{s})$ [24] as a measure of closeness to ID data. The Mahalanobis distance is computed as follows:

$$\mathrm{M}(\mathbf{s}) = \max_c \left( -(h_{\text{text}}(\mathbf{s}) - \hat{\mu}_c)^\top \hat{\boldsymbol{\Sigma}}^{-1} (h_{\text{text}}(\mathbf{s}) - \hat{\mu}_c) \right), \tag{2}$$

where $\hat{\mu}$ and $\hat{\Sigma}$ denote class-wise mean and variance of the training set. $\hat{\mu}$ and $\hat{\Sigma}$ are derived using the following equations:

$$\hat{\mu}_c = \frac{1}{N_c} \sum_{i:y_i=c} h_{\text{image}}(\mathbf{x}_i), \quad \hat{\Sigma} = \frac{1}{N} \sum_c \sum_{i:y_i=c} (h_{\text{image}}(\mathbf{x}_i) - \hat{\mu}_c)(h_{\text{image}}(\mathbf{x}_i) - \hat{\mu}_c)^\top, \quad (3)$$

where $h_{\{\}}(\cdot)$ refers to the CLIP image or text encoder. We only consider the top-$p$% of the data with the largest $\mathrm{M}(\mathbf{s})$.

## 4.2 Optimizing ID embeddings with textual outlier

We train our classifier using the loss as defined below:

$$\mathcal{L}(f) = \mathbb{E}_{(\mathbf{x},y) \sim \mathcal{D}_{ID}} \left[ \mathcal{L}_{CE} \left( f(h_{\text{image}}(\mathbf{x})), y \right) \right] + \lambda \mathbb{E}_{\mathbf{s} \sim \mathcal{D}_{out}^{OE}} \left[ \mathcal{L}_{OE}(f(h_{\text{text}}(\mathbf{s})) \right], \quad (4)$$

where the left term is cross-entropy loss and the right term is Kullback-Leibler divergence to the uniform distribution, which can be expressed as $-\frac{1}{C} \sum_c \texttt{softmax}_c f(\cdot)$. $\mathcal{D}_{out}^{OE}$ refers to the generated textual outliers. Our loss aims to increase the classification performance on train data and reduce the confidence on $\mathcal{D}_{out}^{OE}$. We add Gaussian noise to the textual outliers to reduce the modality gap [36].

## 4.3 Test time OoD detection

When testing, we use Energy score [29] for OoD detection $S(\mathbf{x}, f) = -E(\mathbf{x}; f)$, where

$$E(\mathbf{x}; f) = -T \cdot \log \sum_{i:y_i=c}^{C} e^{f_i(x)/T}. \quad (5)$$

Energy score is known to be less susceptible to the overconfident issue. It is worth mentioning that the sign of the energy function, denoted as $E(\mathbf{x})$, is reversed to align with the convention where samples with higher scores are classified as ID, while samples with lower scores are classified as OoD. Our method can be used with any OoD score that utilizes output probabilities.

# 5   Experiments

In this section, we conduct extensive experiments to showcase the effectiveness of our novel technique for textual outlier exposure. Through comprehensive experiments, we evaluate and investigate our approach from three main perspectives: (1) comparison of our method with existing methods in outlier exposure, (2) comparison of performance among proposed textual outliers, and (3) assessment of the proposed textual outlier exposure in challenging OoD detection scenarios.

**Datasets.** We use the large-scale ImageNet-1K [6] OoD detection benchmark proposed by Huang *et al.* [20]. We conduct experiments on four OoD test datasets: subsets of iNaturalist [47], SUN [52], Places [58], Texture [4]. The categories of OoD datasets are disjoint from those of the ID dataset.

**Experimental setting.** The CLIP model used in this paper is adapted from OpenAI's public repository, with ViT-B/16 serving as the default vision and language backbone. For word-level textual outliers, the BERT [7] large model with 24 layers and a hidden size of 1024 from huggingface [51] is used. The model is trained on the MS-COCO [28] dataset using the Adam optimizer [22] with a fixed learning rate of $10^{-5}$ for 100 epochs. A simple teacher-forcing algorithm is employed to retrieve text during the training process. To generate description-level textual outliers, the GPT-3-text-davinci-002 model is used. For caption-level textual outliers, BLIP-2 opt 2.7b from huggingface is used. The Adam optimizer is used to train our linear classifier for OoD detection, and batch size, learning rate, and other hyperparameters are tuned on the validation set. We use 0.5 for $\lambda$ in our training loss, and batch size for both ID and train time outlier is 32. The temparature value $T$ for Energy score is set to 1, as per the original paper. Model checkpoints with the highest validation accuracy are evaluated on the test set. We employ a value of 30 for $k$, which is used in word-level outliers filtering, and a value of 25 for $\delta$. Furthermore, a filtering ratio of 15 is utilized as the $p$ for caption-level outlier analysis. The ablation study examining these parameters can be found in the Appendix B.

Table 5: Comparison of proposed textual outliers and competitive baselines from visual outlier exposure on the ImageNet-1K dataset. The best result in each column is in bold.

| OoD Data | iNaturalist | | SUN | | Places | | Texture | | | Average | |
| --- | --- | --- | --- | --- | --- | --- | --- | --- | --- | --- | --- |
| Metrics | FPR95 | AUROC | FPR95 | AUROC | FPR95 | AUROC | FPR95 | AUROC | ID Acc | FPR95 | AUROC |
| Visual OE | | | | | | | | | | | |
| OE | 78.31 | 75.23 | 80.10 | 76.55 | 70.41 | 81.78 | 66.38 | 82.04 | 75.51 | 73.80 | 78.90 |
| CSI | 75.85 | 82.63 | 90.62 | 47.83 | 94.90 | 44.62 | 85.85 | 87.11 | 74.27 | 86.80 | 65.54 |
| SSD+ | 59.60 | 85.54 | 75.62 | 73.80 | 83.60 | 68.11 | 39.40 | 82.40 | **78.80** | 64.55 | 77.46 |
| MixOE | 80.51 | 74.30 | 74.62 | 79.81 | 84.33 | 69.20 | 58.00 | 85.83 | 74.62 | 74.36 | 77.28 |
| VOS | 94.83 | 57.69 | 98.72 | 38.50 | 87.75 | 65.65 | 70.20 | 83.62 | 74.43 | 87.87 | 61.36 |
| DOE | 55.87 | 85.98 | 80.94 | 76.26 | 67.84 | 83.05 | **34.67** | **88.90** | 75.50 | 59.83 | 83.54 |
| Textual OE | | | | | | | | | | | |
| None | 84.74 | 74.02 | 92.52 | 63.85 | 99.21 | 41.72 | 99.01 | 27.30 | 72.43 | 93.87 | 51.72 |
| Word | 31.72 | 94.56 | 58.76 | 86.73 | 67.68 | 83.51 | 80.12 | 77.10 | 76.60 | 59.57 | 85.47 |
| Desc. | 33.12 | 94.28 | 54.72 | 87.70 | 72.47 | 81.82 | 86.03 | 70.72 | 77.48 | 61.58 | 83.63 |
| **Caption** | **29.61** | **94.74** | **57.12** | **87.34** | **66.82** | **83.71** | 79.29 | 77.76 | 76.00 | **58.21** | **85.88** |

## 5.1 Comparison with competitive baseline approaches

**Textual outliers outperform previous advanced methods.** To compare the effectiveness of textual outlier exposure method with other outlier exposure methods, we consider the following baselines: OE [18], CSI [45], SSD+ [41], MixOE [55], VOS [8], and DOE [50]. The performances of compared methods in Table 5 are borrowed from those reported in Wang *et al.* [50]. All of these baseline methods utilize CNN architectures, specifically ResNet [16] or WideResNet [54].

Our textual outlier methods at every level consistently achieve superior OoD detection performance, surpassing competing approaches in terms of AUROC. Caption- and word-level textual outliers succeed at improving the FPR95 metric as well. We point out that the caption-level textual outliers reduce FPR95 from 87.87% (VOS) to 58.21% (ours), resulting in a substantial improvement of 29.66%p. This significant performance gap highlights the effectiveness of our textual outlier generation method for model regularization. The results on the ImageNet-100 benchmark and comparison with post-hoc methods can be found in the Appendix A. We include experimental results for CLIP-RN50 and RN50x4 in the Appendix D.4 and D.5 to demonstrate that the observed performance gains do not stem from different architectures, namely ResNet and ViT. The performance comparison among the proposed textual outliers is included in the lower half of Table 5. We also report the results of using CLIP without outlier exposure (None) to demonstrate that the performance improvement is not simply due to the inductive bias of the CLIP embedding space. Among the three levels of textual outliers, the caption-level textual outliers demonstrate superior performance on average.

Based on these results, we can deduce that textual outliers must meet the characteristics summarized below to function effectively for outlier exposure. While the near-distribution criterion is in line with previous insights from the outlier exposure literature, the other two are unique to textual outliers.

- **Near-Distribution:** Textual outliers should be situated near the boundary samples of the training distribution, as several approaches in visual outlier exposure have emphasized [23, 33, 3]. This allows the neural network to compactly estimate the decision boundary between ID and OoD data [8].

- **Descriptiveness:** Textual outliers should possess rich linguistic semantics to include comprehensive and diverse explanations of OoD data. Word-level textual outliers, which are relatively terse, often fall short of the other two forms of textual outliers in this aspect.

- **Inclusion of Visual Semantics:** Textual outliers should incorporate visual elements, such that generated textual outliers do not deviate far from visual boundary samples of ID data.

The characteristics satisfied by textual outliers at each level are detailed in the Appendix F.3.

Table 6: Effectiveness of textual outliers exposure in hard OoD situation, a more challenging OoD detection scenario. Proposed textual outliers are compared against strong baseline approaches on three different ID/OOD dataset combinations. The best result in each column is in bold.

| Dataset | ID OoD | ImageNet10 ImageNet20 | | ImageNet20 ImageNet10 | | CUB100 CUB100 | |
|---|---|---|---|---|---|---|---|
| | | FPR | AUROC | FPR | AUROC | FPR | AUROC |
| Visual OE | OE | 93.20 | 54.51 | 89.80 | 57.63 | 87.20 | 60.70 |
| | VOS | 91.00 | 60.42 | 93.50 | 44.38 | 94.79 | 49.25 |
| Textual OE | Word | 4.60 | 98.92 | 9.00 | 98.39 | 80.34 | 71.62 |
| | **Desc.** | **3.60** | **99.02** | **7.20** | **98.69** | **82.56** | **72.33** |
| | Caption | 5.30 | 98.85 | 8.60 | 98.43 | 85.92 | 64.79 |

## 5.2 Textual outliers in hard OoD situation

**Description-level textual outliers outperforms in hard OoD.** Furthermore, we explore the effectiveness of textual outliers in handling challenging OoD inputs. In these scenarios, the OoD samples closely resemble the ID samples in terms of their semantic content. To evaluate the performance of OoD detection in such a semantically hard scenario, we first utilize the ImageNet10 *vs.* ImageNet20 benchmark proposed by Ming *et al.* [34]. This task involves distinguishing between high-resolution images with semantically similar categories, such as dog *vs.* wolf. Our experimental results, shown in Table 6 (first and second columns), demonstrate the superiority of our textual outliers over OE, achieving an 89.6%p improvement in FPR95 for ImageNet10 (ID) *vs.* ImageNet20 (OoD), and a 82.6%p improvement when ID and OoD data are switched.

Additionally, we conduct experiments on the CUB dataset [49] benchmark, which Vaze *et al.* [48] proposed (third column in Table 6). This benchmark is designed to discriminate between ID and OoD samples that possess numerous shared attributes, thereby introducing difficulties in accurately identifying OoD instances. As shown in Table 6, our description-based textual outliers outperform the auxiliary dataset and synthesis methods. While descriptions are known to improve classification performance, they can also be effective outliers in situations where class names are not considered. The class labels for the hard OoD benchmarks are provided in the Appendix F.2.

## 6 Related work

**OoD detection.** A considerable body of research has been dedicated to designing scoring functions for detecting samples that lie outside the training categories. These methods include Maximum Softmax Probability [17], ODIN score [27], Mahalanobis-based score [24], Energy score [29], gradient-based score [19], and non-parametric KNN score [44]. While there have been studies proposing OoD scores using CLIP's textual information [32, 12, 10], none of them have incorporated textual information into outlier exposure during training. An alternative approach to address the out-of-distribution (OoD) detection problem is through training-time regularization, as discussed in previous research [23, 18]. These methods encourage models to provide predictions with lower confidence or higher energies, aiming to improve OoD detection. However, most of these approaches rely on the availability of an auxiliary outlier dataset. To overcome this limitation, recent studies have explored the synthesis of virtual outliers in the feature space, as seen in works such as [8, 46]. This approach aims to generate synthetic outliers without the need for an additional dataset. Our proposed textual outliers differ from previous methods in that they are defined in the input space and do not rely on an auxiliary dataset.

**Multi-modal models.** The adoption of pre-trained VLMs in multi-modal tasks has attracted considerable interest and yielded impressive results. Prominent among these approaches are dual-stream models such as CLIP [37], ALIGN [21], and FILIP [53]. These models utilize distinct encoders for text and image data and optimize them through contrastive objectives to align semantically similar features across heterogeneous modalities. The success of CLIP-like models inspired numerous subsequent investigations [26, 56, 59], aiming to enhance data efficiency and adaptability for various downstream tasks. Recent progress in this field, exemplified by innovations such as DALL-E 2 [38], ClipCap [35], and related studies [5, 13], has pushed the boundaries of exploration in multi-modal contrastive representation spaces, uncovering their untapped potential. To the best of our knowledge, this is the first time that joint embedding has been utilized for outlier exposure.

# 7 Conclusion

In this paper, we propose a novel outlier framework, transitioning from the traditional single-modal paradigm to a multi-modal regime. Unlike existing methods that rely on image data, we leverage textual outliers during the training process. We analyze the advantages of textual outliers compared to visual outliers and propose three levels of textual outliers. Our proposed textual outliers significantly enhance the neural network's capability to distinguish between ID and OoD data, leading to superior OoD detection performance while maintaining the classification performance on ID data. Furthermore, our analysis of each level of outliers provides valuable insights into the characteristics of effective textual outliers. We anticipate that our work will serve as a source of inspiration for future research on OoD detection methods that leverage textual outlier exposure. **Limitations.** Our proposed textual outlier method involves a heuristic selection process to refine outputs of generative models. In future work, it is necessary to reduce the dependence on heuristic methods and explore alternative approaches.

## Acknowledgments

This work was supported by the National Research Foundation of Korea (NRF) grant funded by the Korea government (MSIT) (2022R1A3B1077720 and 2022R1A5A708390811), Institute of Information & Communications Technology Planning & Evaluation (IITP) grants funded by the Korea government (MSIT) [2021-0-01343: Artificial Intelligence Graduate School Program (Seoul National University) and 2022-0-00959] and the BK21 FOUR program of the Education and Research Program for Future ICT Pioneers, Seoul National University in 2023.

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
