# On the Powerfulness of Textual Outlier Exposure for Visual OoD Detection (Appendix)

## A    Additional experimental results

This section presents more comprehensive experimental results. We employ the CLIP ViT-B/32 for Section A.1 and A.2, CLIP ViT-B/16 for Section A.3.

### A.1    Comparison with post-hoc methods

We also compare the performance of our textual outlier method with post-hoc approaches, which are another prominent approach in OoD detection. We conducted comparisons with six widely used and recently proposed methods known for their detection performance (MSP [4], ODIN [8], Mahalanobis [7], Energy [10], ReAct [14], KNN [15]). All advanced baseline methods follow the original paper's settings. Among these methods, our textual outlier approach demonstrate the best performance, further emphasizing its effectiveness as demonstrated in Table 6.

### A.2    ImageNet100 results

To evaluate the performance of our proposed method on another benchmark dataset (ImageNet100), we compare our textual outliers with other advanced outlier exposure methods, such as OE [5], VOS [3], and DOE [19]. As shown in Table 7, our textual outliers demonstrated superior performance compared to these existing methods, further highlighting the effectiveness of our method.

### A.3    Error bar

We conducted five repetitions of training our method on ImageNet-1K. The reported results in Table 1 include the mean values and standard deviations of the performance measures. We only report the performance on the caption-level textual outliers that achieves the best performance.

Table 1: Results on the mean and standard deviations after 5 runs.

|         | FPR95          | AUROC          |
|---------|----------------|----------------|
| Caption | $57.93 \pm 1.26$ | $85.87 \pm 0.42$ |

## B    Hyperparameters for filtering Textual outliers

We determine the optimal hyperparameters for textual outliers through a comprehensive ablation study. Our hyperparameter search for filtering textual outliers is also conducted using the CLIP ViT-B/32 model.

### B.1    Ablation study on filtering ratio of caption-level

In our ablation study for caption-level textual outliers, we investigate the impact of the top percentage of Mahalanobis distance values used in the filtering process to identify textual outliers. We test different values of $p$, including 0.1, 0.15, and 0.2. The results reveal that when $p$ was set to 0.15, it yields the best performance in terms of OoD detection as shown in Table 12.

37th Conference on Neural Information Processing Systems (NeurIPS 2023).

## B.2 Ablation study on filtering ratio of word-level

Similar to caption-level textual outliers, word-level textual outliers also incorporate a filtering parameter. In order to determine the optimal filtering ratio, we perform an ablation study to analyze its impact on the overall performance. As shown in Table 13, we found that the performance reaches its peak when words with similarity values between the multi-head attention of the BERT model and the image embeddings fall within the range of the top 30 to 55. Additionally, we investigate the influence of the sample size on performance by testing the top 50 as well. However, we observe that including a larger sample size does not lead to any significant improvement in performance.

# C Detailed Exposition of Section 3

## C.1 Additional results for Section 3.1

To further analyze the performance variance with respect to the auxiliary dataset, we conduct additional comparative experiments on ImageNet10 (Table 8) and ImageNet100 (Table 9). In the case of ImageNet100, we use Texture [1] as OoD dataset. Therefore, it is not utilized as an outlier. Across all benchmarks, when utilizing textual outliers, we observe lower performance variance compared to image outliers. We would like to mention that the experiments in Section 3 are conducted using the CLIP ViT-B/32 backbone.

## C.2 Synonyms and descriptions for Section 3.2

**warplane (n04552348).** *synonym*: fighter jet, combat aircraft, bomber, military plane, attack aircraft, interceptor, gunship, reconnaissance plane, aerial warfare platform, strike aircraft. *description*: "large and powerful", "designed for carrying weapons and other military equipment", "typically has a camouflage paint job", "often has a "star" or other symbol on the fuselage to identify the country of origin", "may have a tailfin with a "missile" or other symbol."

**sports car (n04285008).** *synonym*: performance car, roadster, sports car convertible, sporty car, two-seater car, racing car, muscle car, high-performance car, exotic car, luxury sports car. *description*: "a vehicle with two or four doors", "a sleek, aerodynamic design", "a powerful engine", "large wheels and tires", "a spoiler or other performance-enhancing features", "a stylish interior."

**brambling bird (n01530575).** *synonym*: Eurasian brambling, mountain finch, bramble finch, Bramble bird, Common brambling, Fringilla montifringilla, Northern mountain finch, Red-winged brambling, Winter finch, Rustic bunting. *description*: "a small, sparrow-like bird", "brown and white plumage", "a black head with a white stripe above the eye", "a black bill", "a forked tail", "yellow legs."

**Siamese cat (n02123597).** *synonym*: Thai cat, Royal Siamese cat, Traditional Siamese cat, Old-style Siamese cat, Applehead Siamese cat, Seal point cat, Chocolate point cat, Blue point cat, Lilac point cat, Flame point cat. *description*: "blue eyes", "pointy ears", "long, slender body", "short fur", "light-colored fur with dark points on the face, ears, legs, and tail."

**antelope (n02422699).** *synonym*: gazelle, deer, ibex, pronghorn, kudu, impala, springbok, oryx, sable , wildebeest. *description*: "four-limbed mammal", "reddish-brown or tan coat", "black stripes on the hindquarters", "long, black tail with a white tuft at the end", "black horns", "large, dark eyes."

**Swiss mountain dog (n02107574).** *synonym*: Bernese Mountain Dog, Appenzeller Sennenhund, Entlebucher Mountain Dog, Greater Swiss Mountain Dog, Sennenhund-type dog, Swiss cattle dog, Alpine Mastiff, Berner Sennenhund, Swissy, Berner. *description*: "large, muscular body", "thick, double coat of fur", "black, brown, or white with black markings", "long head with a square muzzle", "dark eyes", "triangular ears", "strong, straight legs", "large, round feet", "long tail."

**bull frog (n01641577).** *synonym*: American bullfrog, Rana catesbeiana, bull toad, giant frog, North American bullfrog, green frog, pond frog, bull croaker, lake frog, water frog. *description*: "large size", "green or brown body", "dark spots on the body", "webbed feet", "long hind legs for jumping", "large eardrums", "long tongue"

**garbage truck (n03417042).** *synonym*: refuse truck, waste collection vehicle, dustbin lorry, trash truck, rubbish truck, garbage collector, waste truck, compactor truck, sanitation truck, bin lorry.

*description*: "large, boxy vehicle", "brightly colored", ""Garbage Truck" or "Sanitation" Truck markings", "rear loading door", "hydraulic lift arm", "large tires", "often has a rear-view camera."

**horse (n02389026).** *synonym*: equine, mare, stallion, gelding, colt, filly, mustang, pony, steed, nag. *description*: "reddish-brown coat", "white markings on the face and legs", "black mane and tail", "muscular body", "long head", "short, erect ears", "large eyes."

**container ship (n03095699).** *synonym*: cargo ship, freighter, container vessel, box ship, container carrier, containerized freighter, container barge, containerized cargo ship, container feeder ship container liner. *description*: "large vessel", "blue or grey", "white superstructure", "stacks of containers on deck", "cranes for loading and unloading containers", "lifeboats."

# D    Ablation Study

Extensive ablation studies are conducted to validate the strategies employed in constructing our method. Any experiments in this section are also performed using the CLIP ViT-B/32 model.

## D.1    Comparison with image embedding

For the case of caption-level textual outliers, we employ a combination of images and a captioning model to generate captions. Subsequently, the Mahalanobis distance is computed based on the acquired captions to identify textual outliers. In order to evaluate the performance of the caption-level textual outliers, we also explore an alternative approach that involves using image embeddings directly to calculate the Mahalanobis distance, without incorporating the captioning process. Our experimental results demonstrate that utilizing captioning and deriving textual outliers from the generated captions yields superior performance compared to using image embeddings as outliers. This improvement in performance is consistently observed across all evaluated OoD datasets as shown in Figure 1.

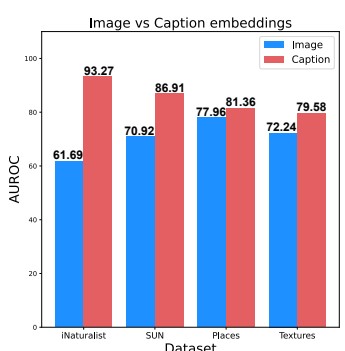

Figure 1: Comparison of AUROC achieved by using caption-level textual outlier and image embedding

## D.2    The impact of prompt ensembling

For word-level and description-level textual outliers, we utilize prompt ensembling (referred to as "pe") to enhance their effectiveness. To assess the impact of prompt ensembling, we conduct an ablation study comparing it with the case without prompt ensembling, where only the prompt "a photo of" is used. As shown in Table 2, prompt ensembling yields performance improvements.

Table 2: Ablation on prompt ensembling to word- and desc-level textual outliers.

|       |          | Average |        |
|-------|----------|---------|--------|
|       |          | FPR95   | AUROC  |
| Word  | w/o pe   | 64.73   | 83.31  |
|       | **w/ pe**| **59.11** | **85.45** |
| Desc. | w/o pe   | 70.60   | 81.08  |
|       | **w/ pe**| **65.40** | **83.62** |

## D.3    The impact of noise

As discussed in various literature, there exists a modality gap between image and text embeddings [9, 12]. To enhance the efficacy of textual outliers in visual OoD detection tasks, we explore approaches to minimize the modality gap. Previous studies have demonstrated that introducing noise to text embeddings can effectively reduce the modality gap between the textual and visual domains [12]. We follow this strategy and add Gaussian noise with zero mean and variance $\epsilon^2$ to the textual outliers we define

Table 3: Ablation on noise injection to text embeddings.

|           |        | Avg    |        |
|-----------|--------|--------|--------|
|           | ID Acc | FPR95  | AUROC  |
| w/o noise | 71.07  | 64.14  | 83.26  |
| **w/ noise** | 72.71 | **58.35** | **85.53** |

$$h_{\text{text}}(\mathbf{s}) = h_{\text{text}}(\mathbf{s}) + n \sim \mathcal{N}(0, \epsilon^2) \tag{1}$$

where $\epsilon^2$ is 0.016. The ablation study demonstrates the effectiveness of adding noise as shown in Table 3. The results are obtained from caption-level outliers and ID dataset is ImageNet-1K.

### D.4 Ablation study on CLIP model scale

We evaluate the performance of the CLIP [13] as the size of the model varies. Using ViT-B/32 as the baseline, we employ a larger-scale model, ViT-L/14, and a smaller-scale model, RN50, for comparison. We observe that as the scale of the CLIP model increased, the performance also improved as shown in Table 10. This indicates that utilizing a larger scale for the CLIP model has a positive impact on its ability to detect outliers and improve overall performance in OoD detection tasks. Remarkably, our textual outlier exhibited comparable performance levels (AUROC) even with a significantly smaller model size (176M vs 77M) as shown in Table 4. The results for small-scale experiments are obtained from description-level textual outliers.

Table 4: Ablation on model capacity. ID dataset is ImageNet-1K.

|          | Parameter size | Avg   |
| -------- | -------------- | ----- |
| ViT-B/32 | 176M           | 83.62 |
| RN50     | 77M            | 82.53 |

### D.5 Ablation study on different backbone architecture

We conducted experiments to compare the performance of our method across two different image encoder architectures, ResNet and ViT, the two architectures offered by CLIP. Importantly, as shown in Table 11, our method consistently produces encouraging outcomes, even when applied to CLIP models built upon the ResNet architecture. We selected RN50x4, designed with a parameter size akin to that of ViT-B/32 (174M vs 176M). The performance analysis reveals a comparable trend between RN50x4 and ViT-B/32, resulting in AUROC scores of 86.43 and 87.55, respectively. For this experiment, we utilized ImageNet1K as the in-distribution dataset. The results are obtained from caption-level textual outliers.

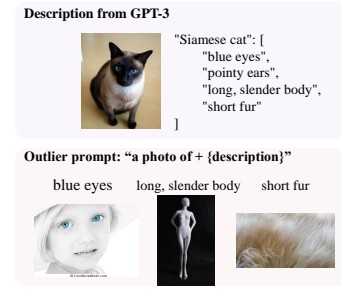

Figure 2: An example of a description-level textual outlier.

## E Detailed Explanation

### E.1 Description-level textual outlier examples

In the context of image retrieval, if we use the description of a Siamese cat as a prompt, such as "a photo of + *short fur*," the CLIP model retrieves images that exhibit substantial deviations from the actual visual characteristics of a Siamese cat. An illustrative example image demonstrating this phenomenon can be found in Figure 2.

### E.2 More textual outlier examples

Here are a few examples of textual outliers for ImageNet. We offer three distinct types of textual outliers for each class.

Class: *Warplane*

- word: "This is a photo of army", "This is a photo of airport", "This is a photo of airliner"
- desc: "a photo of large and powerful", "a photo of designed for carrying weapons and other military equipment", "a photo of typically has a camouflage paint job",
- caption: "a small plane with a window on the side", "a silver airplane", "a plane with a large window on the front", "a plane taking off"

Class: *Greater Swiss Mountain Dog*

- word: "This is a photo of green", "This is a photo of puppy", "This is a photo of pets"
- desc: "a photo of large, fluffy white dog", "a photo of black or brown markings on the face, ears, and tail", "a photo of long, thick coat"
- caption: "two dogs are playing with each other", "a dog with a white face", "a dog with its tongue out"

Table 5: The characteristics fulfilled by textual outliers at each level, along with the sample size of each textual outliers on the ImageNet-1K.

| | Near-Distribution | Descriptiveness | Inclusion of Visual Semantics | Statistics |
|---|---|---|---|---|
| Word | ✓ | | ✓ | 1393 |
| Desc. | ✓ | ✓ | | 5800 |
| Caption | ✓ | ✓ | ✓ | 4561 |

Based on the provided examples, our word-level outliers convey more abstract concepts. Similarly, caption-level outliers mostly contain descriptions of background elements or lack class-specific attributes. Description-level outliers include class-relevant information, but when the class label is omitted, they become very vague and difficult to interpret. In the revised manuscript, we will incorporate additional examples to enhance understanding.

## F   Experimental Details

### F.1   Software and Hardware

All our experiments are implemeted using PyTorch and conducted with NVIDIA Quadro RTX8000 GPU.

### F.2   Datasets

**ImageNet10 and ImageNet20.** We utilize the same ImageNet10 and ImageNet20 which is defined by Ming *et al.* [11].

**ImageNet100.** We use the same 100 classes from ImageNet-1K [2] as defined in [11, 16] to create ImageNet100.

**Fine-grained benchmark (CUB200).** We utilize the fine-grained open set classes defined based on the similarity calculated using attributes of the CUB dataset in the Vaze *et al* [18].

*Known classes*: [150, 70, 34, 178, 199, 131, 129, 147, 134, 11, 26, 93, 95, 121, 123, 99, 149, 167, 18, 31, 69, 198, 116, 158, 126, 17, 5, 179, 111, 163, 184, 81, 174, 42, 53, 89, 77, 55, 23, 48, 43, 44, 56, 28, 193, 143, 0, 176, 84, 15, 38, 154, 141, 190, 172, 124, 189, 19, 80, 157, 12, 9, 79, 30, 94, 67, 197, 97, 168, 137, 119, 76, 98, 88, 40, 106, 171, 87, 166, 186, 27, 51, 144, 135, 161, 64, 177, 7, 146, 61, 50, 162, 133, 82, 39, 74, 72, 91, 196, 136].

*Unknown classes*: 'Easy': [20, 159, 173, 148, 1, 57, 113, 165, 52, 109, 14, 4, 180, 6, 182, 68, 33, 108, 46, 35, 75, 188, 187, 100, 47, 105, 41, 86, 16, 54, 139, 138], 'Medium': [152, 195, 132, 83, 22, 192, 153, 175, 191, 155, 49, 194, 73, 66, 170, 151, 169, 96, 103, 37, 181, 127, 78, 21, 10, 164, 62, 2, 183, 85, 45, 60, 92, 185], 'Hard': [29, 110, 3, 8, 13, 58, 142, 25, 145, 63, 59, 65, 24, 140, 120, 32, 114, 107, 160, 130, 118, 101, 115, 128, 117, 71, 156, 112, 36, 122, 104, 102, 90, 125]

We use every level of unknown classes for OoD dataset in fine-grained benchmark setting.

**OoD datasets.** Huang et al. [6] create a varied selection of subsets from iNaturalist [17], SUN [20], Places [21], and Texture [1] datasets to form large-scale OoD datasets for ImageNet-1K. In these OoD datasets, the classes in the test sets are distinct and do not have any overlap with the classes in ImageNet-1K. We use the same OoD datasets as defined in [6].

### F.3   Textual-outlier statistics

When using $k$=30, $\delta$=25, and $p$=0.15 values on the ImageNet-1K benchmark, the number of textual outlier samples is as follows for each level: 1393 for word-level, 5800 for description-level, and 4561 for caption-level. Along with these statistics, in the Table 5, the characteristics fulfilled by each textual outlier are included.

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

Table 6: Comparison of proposed textual outliers and competitive baselines from post-hoc methods on the ImageNet-1K dataset. The best result in each column is in bold.

| | OoD datasets | | | | | | | | Average | |
| | iNaturalist | | SUN | | Places | | Textures | | | |
| | FPR95 | AUROC | FPR95 | AUROC | FPR95 | AUROC | FPR95 | AUROC | FPR95 | AUROC |
|---|---|---|---|---|---|---|---|---|---|---|
| Post-hoc | | | | | | | | | | |
| MSP | 72.98 | 77.22 | 80.89 | 74.24 | 76.69 | 77.81 | 70.73 | 78.58 | 75.32 | 76.96 |
| ODIN | 63.85 | 77.78 | 89.98 | 61.80 | 88.00 | 67.17 | 67.87 | 77.40 | 77.43 | 71.04 |
| Mahalanobis | 95.90 | 60.56 | 95.42 | 45.33 | 98.90 | 44.65 | 55.80 | 84.60 | 86.50 | 58.78 |
| Energy | 69.10 | 77.39 | 82.36 | 76.08 | 76.15 | 80.23 | 56.97 | 84.32 | 71.14 | 79.50 |
| ReAct | 56.11 | 84.94 | 82.79 | 75.87 | 75.00 | 80.72 | 70.27 | 82.16 | 70.31 | 81.42 |
| KNN | 65.40 | 83.73 | 75.62 | 77.33 | 79.20 | 74.34 | **40.80** | **86.45** | 64.75 | 80.91 |
| Textual OE | | | | | | | | | | |
| Word | **32.65** | 94.42 | 56.68 | **86.96** | 71.06 | 81.65 | 76.08 | 78.80 | 59.11 | 85.45 |
| Desc. | 46.09 | 92.07 | 60.64 | 85.24 | 81.28 | 77.40 | 73.62 | 79.77 | 65.40 | 83.62 |
| **Caption** | 32.91 | **94.55** | **55.68** | 86.59 | **70.54** | **81.51** | 74.29 | 79.48 | **58.35** | **85.53** |

Table 7: Comparison of proposed textual outliers and competitive baselines from visual outlier exposure on the ImageNet100 dataset. The best result in each column is in bold.

| | OoD datasets | | | | | | | | Average | |
| | iNaturalist | | SUN | | Places | | Textures | | | |
| | FPR95 | AUROC | FPR95 | AUROC | FPR95 | AUROC | FPR95 | AUROC | FPR95 | AUROC |
|---|---|---|---|---|---|---|---|---|---|---|
| Visual OE | | | | | | | | | | |
| OE | 90.46 | 59.11 | 86.34 | 60.54 | 75.28 | 69.39 | 92.72 | 64.02 | 86.20 | 63.26 |
| VOS | 97.50 | 47.60 | 90.00 | 65.46 | 86.10 | 68.66 | 86.60 | 63.55 | 90.05 | 61.31 |
| DOE | 82.80 | 66.06 | 85.20 | 61.77 | 93.30 | 58.78 | 92.70 | 61.23 | 88.50 | 61.96 |
| Textual OE | | | | | | | | | | |
| Word | 26.78 | 96.09 | 25.80 | **95.45** | **32.32** | **93.84** | 37.93 | 93.81 | 30.70 | **94.79** |
| Desc. | **16.50** | **97.19** | 26.75 | 94.94 | 35.97 | 93.04 | 38.17 | 93.88 | **29.34** | 94.76 |
| Caption | 35.75 | 95.26 | **25.76** | 95.05 | 35.60 | 93.17 | **36.99** | **94.08** | 33.52 | 94.39 |

Table 8: Results of using images *vs.* texts (class labels) of auxiliary datasets as outliers. We compare the results of utilizing 3 different auxiliary datasets. In the testing phase, we use ImageNet10 and ImageNet20 as ID and OoD datasets, respectively. The best result in each column is in bold.

| | Auxiliary datasets | | | | | | | | ID Acc | Average | |
| | Textures | | Places | | SUN | | iNaturalist | | | | |
| Outliers | FPR | AUROC | FPR | AUROC | FPR | AUROC | FPR | AUROC | | FPR | AUROC |
|---|---|---|---|---|---|---|---|---|---|---|---|
| None | - | - | - | - | - | - | - | - | 99.6 | 10.70 | 97.86 |
| Image | **3.90** | **98.79** | **5.50** | **98.65** | 10.70 | 97.64 | **5.50** | 97.80 | 99.6 | **6.40** $\pm$ 2.96 | **98.22** $\pm$ 0.58 |
| Text | 9.10 | 98.38 | 9.50 | 98.03 | **7.80** | **98.18** | 11.30 | **97.98** | **99.8** | 9.42 $\pm$ 1.44 | 98.14 $\pm$ 0.17 |

Table 9: Results of using images *vs.* texts (class labels) of auxiliary datasets as outliers. We compare the results of utilizing 3 different auxiliary datasets. In the testing phase, we use ImageNet100 and Textures as ID and OoD datasets, respectively. The best result in each column is in bold.

| | Auxiliary datasets | | | | | | ID Acc | Average | |
| | Places | | SUN | | iNaturalist | | | | |
| Outliers | FPR | AUROC | FPR | AUROC | FPR | AUROC | | FPR | AUROC |
|---|---|---|---|---|---|---|---|---|---|
| None | - | - | - | - | - | - | 90.6 | 28.56 | 94.89 |
| Image | 42.89 | 93.22 | 38.65 | 94.06 | **19.36** | 96.58 | **91.2** | 33.63 $\pm$ 12.54 | 94.62 $\pm$ 1.74 |
| Text | **20.60** | **97.26** | **22.60** | **97.08** | 22.20 | **96.94** | 90.0 | **21.25** $\pm$ 0.86 | **97.13** $\pm$ 0.10 |

Table 10: Ablation on model capacity. ID dataset is ImageNet-1K.

| | | OoD datasets | | | | | | | | | Average | |
| | | iNaturalist | | SUN | | Places | | Textures | | | | |
| | | FPR95 | AUROC | FPR95 | AUROC | FPR95 | AUROC | FPR95 | AUROC | ID Acc | FPR95 | AUROC |
|---|---|---|---|---|---|---|---|---|---|---|---|---|
| Word | ViT-B/32 | 32.65 | 94.42 | 56.68 | 86.96 | 71.06 | 81.65 | 76.08 | 78.80 | 72.15 | 59.12 | 85.45 |
| | **ViT-L/14** | 26.83 | 95.01 | 45.99 | 89.39 | 52.21 | 87.25 | 50.04 | 87.74 | 80.14 | **43.76** | **89.84** |
| Desc. | ViT-B/32 | 46.09 | 91.79 | 60.64 | 85.24 | 81.28 | 77.40 | 77.36 | 77.59 | 72.66 | 66.34 | 83.00 |
| | **ViT-L/14** | 31.68 | 94.21 | 50.06 | 88.58 | 60.80 | 85.48 | 50.82 | 87.24 | 80.57 | **48.34** | **88.87** |
| Caption | ViT-B/32 | 38.32 | 93.27 | 55.70 | 86.91 | 71.62 | 81.36 | 72.46 | 79.58 | 72.71 | 59.52 | 85.28 |
| | **ViT-L/14** | 20.19 | 95.95 | 46.17 | 89.72 | 52.23 | 87.64 | 50.62 | 87.61 | 80.12 | **42.30** | **90.23** |

Table 11: Ablation on model backbone architectures for caption-level textual outliers. ID dataset is ImageNet-1K.

| | OoD datasets | | | | | | | | | | |
| | iNaturalist | | SUN | | Places | | Textures | | | Average | |
| | FPR95 | AUROC | FPR95 | AUROC | FPR95 | AUROC | FPR95 | AUROC | ID Acc | FPR95 | AUROC |
|---|---|---|---|---|---|---|---|---|---|---|---|
| ViT-B/32 | 32.92 | 94.55 | 55.68 | 86.59 | 70.54 | 81.51 | 74.29 | 79.48 | 71.18 | 58.35 | 87.55 |
| RN50x4 | 51.53 | 91.69 | 64.13 | 85.19 | 66.29 | 82.93 | 57.93 | 85.94 | 73.31 | 59.97 | 86.43 |

Table 12: Ablation on filtering ratio $p$ of caption-level textual outliers.

| | OoD datasets | | | | | | | | | |
| | iNaturalist | | SUN | | Places | | Textures | | Average | |
| ratio | FPR95 | AUROC | FPR95 | AUROC | FPR95 | AUROC | FPR95 | AUROC | FPR95 | AUROC |
|---|---|---|---|---|---|---|---|---|---|---|
| 0.1 | 38.72 | 92.34 | 58.78 | 85.76 | 74.94 | 79.9 | 71.76 | 79.67 | 61.05 | 84.41 |
| **0.15** | 38.32 | 93.27 | 55.7 | 86.91 | 71.62 | 81.36 | 72.46 | 79.58 | **59.52** | **85.28** |
| 0.2 | 41.15 | 92.48 | 65.42 | 83.22 | 80.64 | 77.85 | 74.98 | 79.22 | 65.54 | 83.19 |

Table 13: Ablation on filtering ratio $k$ and $\delta$ of word-level textual outliers.

| | OoD datasets | | | | | | | | | |
| | iNaturalist | | SUN | | Places | | Textures | | Average | |
| $k : k+\delta$ | FPR95 | AUROC | FPR95 | AUROC | FPR95 | AUROC | FPR95 | AUROC | FPR95 | AUROC |
|---|---|---|---|---|---|---|---|---|---|---|
| 1 : 25 | 52.40 | 89.55 | 70.49 | 80.76 | 87.89 | 71.70 | 72.96 | 79.60 | 70.93 | 80.40 |
| 20 : 45 | 42.53 | 92.91 | 66.21 | 83.56 | 84.06 | 76.44 | 73.99 | 79.08 | 66.69 | 82.99 |
| **30 : 55** | 43.05 | 92.49 | 62.27 | 84.33 | 78.4 | 78.00 | 75.23 | 78.44 | **64.73** | **83.31** |
| 40 : 65 | 46.03 | 91.72 | 65.32 | 83.89 | 80.06 | 77.86 | 72.54 | 79.38 | 65.98 | 83.21 |
| 1 : 50 | 44.78 | 91.06 | 62.84 | 84.46 | 79.91 | 77.65 | 75.74 | 78.90 | 65.81 | 83.01 |