# OpenReview forum: "On the Powerfulness of Textual Outlier Exposure for Visual OoD Detection"
_NeurIPS.cc/2023/Conference — NeurIPS 2023 poster_

### Official Review · Reviewer_yebH · 2023-07-02

**Soundness:** 3 good
**Presentation:** 2 fair
**Contribution:** 3 good
**Rating:** 6
**Confidence:** 5

**Summary:**

The paper notices that while outlier exposure has shown promising potential in improving OoD detection performance, all previous studies on outlier exposure have been limited to utilizing visual outliers. The paper uncovers the benefits of using textual outliers by replacing real or virtual outliers in the image domain with textual equivalents. Then, it proposes various ways of generating preferable textual outliers. The extensive experiments demonstrate that generated textual outliers achieve competitive performance on large-scale OoD and hard OoD benchmarks.

**Strengths:**

1. The paper is written well and is easy to understand.
2. The studied problem is very important. section 3 is quite interesting.
3. The results seem to outperform state-of-the-art.


**Weaknesses:**

1. I am curious about section 4.1.1 and section 4.1.3, why do the authors choose to use the images from the validation set of the ID dataset for generation?
2. It is not clear that if the class labels of the outlier dataset is missing, how can we generation description-level supervision in section 4.1.2?
3. I am curious about the performance of the method without the large-scale pretrained models (CLIP) as the classification backbone. The current approach seems to be highly model-specific, which hinders its general usage.
4. The comparison with a reasonable baseline, i.e., NPOS [1] is missing.

[1] Leitian Tao, Xuefeng Du, Jerry Zhu, and Yixuan Li. Non-parametric outlier synthesis. In The Eleventh International Conference on Learning Representations, 2023.

**Questions:**

see above

**Limitations:**

yes

---

> ### Author Rebuttal · Authors · 2023-08-08
>
> Thank you for all of the constructive feedback and suggestions. In particular, we appreciate your recognition of the novelty and efficiency of our work and the insightfulness of our analyses. Here, we show additional experimental results to address your concerns.
>
>
> > W1. Why do the authors choose to use the images from the validation set of the ID dataset for generation?
>
> Due to computational constraints, utilizing the entire training set for generating textual outliers proves time-consuming. Our findings demonstrate that employing only a validation set yields sufficiently favorable performance.
>
> > W2. How can we generation description-level supervision in section 4.1.2?
>
> Our method operates without the need for class labels from the outlier dataset. Instead, we leverage the class labels from the in-distribution data to generate textual outliers, excluding the class label from the descriptions. Descriptions without class labels can be identified as outliers. Illustrative examples can be found in Appendix C, Figure 2.
>
> > W3. I am curious about the performance of the method without the large-scale pretrained models (CLIP) as the classification backbone.
>
> Our method is designed to leverage the power of a joint embedding space that contains both visual and textual embeddings. By definition, such an embedding space is only available in vision-language models, and thus, our method is contingent on the use of VLM models. Regarding your concern with the scalability of our method to smaller models, it's worth noting that CLIP offers a smaller model option, such as ResNet50. Our method still demonstrates comparable performance even on models of reduced size (176M vs 77M). For this experiment, we utilized ImageNet1K as the in-distribution dataset. The results are obtained from description-level textual outliers.
>
> |  | **parameter size** | **iNaturalist** | **SUN** | **Places** | **Textures** | **AVG** |
> |----------|:------------------:|:---------------:|:-------:|:----------:|:------------:|:-------:|
> | ViT-B/32 | 176M               | 92.07           | 85.24   | 77.40       | 79.77        | 83.62   |
> | RN50     | 77M                | 91.01           | 82.44   | 74.95      | 81.70        | 82.52   |
>
> In the revised manuscript, we will incorporate these results to offer a more comprehensive set of experiments.
>
> > W4. The comparison with a reasonable baseline, i.e., NPOS [1] is missing.
>
> Following the suggestion of the reviewer, we experimented with NPOS and the results are as follows.
>
> |  | **iNaturalist** | **SUN** | **Places** | **Textures** | **AVG** |
> |------|:-----------------:|:---------:|:------------:|:--------------:|:---------:|
> | NPOS | 99.14           | 98.06   | 97.69      | 98.19        | 98.27   |
> | ours | 98.86           | 97.93   | 98.29      | 98.48        | **98.39**   |
>
> We employed the ImageNet100 dataset as the in-distribution data and utilized CLIP-L/14, which incorporates a ViT-L/14 Transformer as the image encoder for both approaches. The outcome for our method is derived from caption-level outliers. The average AUROC values across all four OoD datasets highlight that our textual outlier approach outperforms NPOS. In the revised manuscript, we will incorporate these results to offer a more comprehensive set of experiments.

---

### Official Review · Reviewer_JSZe · 2023-07-04

**Soundness:** 3 good
**Presentation:** 4 excellent
**Contribution:** 3 good
**Rating:** 7
**Confidence:** 3

**Summary:**

The paper proposes a new method that takes the textual outliers to help the model better detect out-of-distribution samples. Specifically, they build the pipeline on top of CLIP models with a classifier and use different representations of the text sample to synthesize outliers in the CLIP space, then use them to train the model with ood awareness. The authors provide extensive experiments and analysis of the method.

**Strengths:**

I think the paper is well-written, and the paper is in a great flow. Figures and texts are clear and easy to follow. The authors also give a good rationale and background of the problem.

The authors approach the OOD problem with the textual outliers and the help of CLIP fundamental models, which is a novel and plausible method.

The authors provide extensive experiments and detailed analysis of the problem and extensively worked on different types of textual outliers.


**Weaknesses:**

The authors build the classifiers on top of CLIP, which is a large model with 151 Million parameters (~88 Million in the Image encoder) and implies a large hidden feature space. Comparing this with previous methods may lead to unfair comparisons.

The use of CLIP may limit this to other fields like object detection/segmentations.

**Questions:**

How can you prove the ood robustness is come from your method but not from CLIP itself?

In the appendix, only the larger CLIP model is studied. However,  in the open-sourced CLIP API, they provided smaller models. I wonder how much impact it would have on the metrics as the CLIP shrinks.


**Limitations:**

The method seems only experimented with classifiers. The appliance of the CLIP Image encoder makes it hard to extend to other problems like Object Detection, semantic segmentations, .etc.

---

> ### Author Rebuttal · Authors · 2023-08-08
>
> Thank you for acknowledging the novelty of our method and the extensive experimental results. Here, we show various newly-conducted experimental results to further verify the effectiveness of our method.
>
> > Q1 (W1). How can you prove the ood robustness is come from your method but not from CLIP itself?
>
> In order to demonstrate that the success of our method cannot be attributed solely to the use of CLIP, we have provided results of using CLIP without outlier exposure (labeled as 'None' in Table 5) to substantiate that the performance enhancement is not merely a result of the inherent bias within the CLIP embedding space. Notably, without the integration of textual outliers, the AUROC score is only 51.72, whereas our caption-level textual outliers achieve an AUROC score of 85.53. This disparity underscores that CLIP, without supplementary techniques tailored for OoD detection, yields significantly lower AUROC scores, further attesting that the success of textual outliers is not merely a byproduct of the CLIP embedding space.
>
> NPOS [1] is another work that leverages CLIP's embedding space. NPOS synthesizes outliers from in-distribution data and fine-tunes the CLIP image encoder using these synthesized examples. In the table below, we compare the performance of NPOS vs. Ours. When compared against NPOS, we employed the ImageNet100 dataset as the in-distribution data and utilized CLIP-L/14, which incorporates a ViT-L/14 Transformer as the image encoder for both approaches. The outcome for our method is derived from caption-level outliers. Our approach achieves slightly superior average AUROC values across all four OoD datasets, all without necessitating resource-intensive fine-tuning steps. This implies that textual outliers are potent tools for advancing visual OoD detection.
>
> |  | **iNaturalist** | **SUN** | **Places** | **Textures** | **AVG** |
> |------|:-----------------:|:---------:|:------------:|:--------------:|:---------:|
> | NPOS | 99.14           | 98.06   | 97.69      | 98.19        | 98.27   |
> | ours | 98.86           | 97.93   | 98.29      | 98.48        | **98.39**   |
>
> In the revised manuscript, we will incorporate these results to offer a more comprehensive set of experiments.
>
> [1] Leitian Tao, Xuefeng Du, Jerry Zhu, and Yixuan Li. Non-parametric outlier synthesis. In The Eleventh International Conference on Learning Representations, 2023.
>
> > W2. The use of CLIP may limit this to other fields like object detection/segmentations.
>
> The primary objective of our paper is to improve OOD detection in the context of classification. However, the CLIP encoder can also be extended to object detection or segmentation tasks. Notably, there are approaches designed for object detection [2] and segmentation [3] that are built upon the CLIP architecture.
>
> [2] Yiwu Zhong, et al. RegionCLIP: Region-based Language-Image Pretraining. CVPR 2022
>
> [3] Chong Zhou, Chen Change Loy, and Bo Dai. Extract Free Dense Labels from CLIP. ECCV 2022
>
> > Q2. I wonder how much impact it would have on the metrics as the CLIP shrinks.
>
> Following the suggestion of the reviewer, we conducted an ablation study using a smaller model (ViT-B/32 vs ResNet50). For this experiment, we utilized ImageNet1K as the in-distribution dataset. Remarkably, our textual outlier exhibited comparable performance levels (AUROC) even with a significantly smaller model size (176M vs 77M). The results are obtained from description-level textual outliers.
>
> |  | **parameter size** | **iNaturalist** | **SUN** | **Places** | **Textures** | **AVG** |
> |----------|:------------------:|:---------------:|:-------:|:----------:|:------------:|:-------:|
> | ViT-B/32 | 176M               | 92.07           | 85.24   | 77.40       | 79.77        | 83.62   |
> | RN50     | 77M                | 91.01           | 82.44   | 74.95      | 81.70        | 82.52   |
>
> In the revised manuscript, we will incorporate these results to offer a more comprehensive set of experiments.

---

> > ### Comment · Reviewer_JSZe · 2023-08-18
> >
> > Thank you for your clarification. I think it's a good paper and hope it finds its way into the conference.

---

### Official Review · Reviewer_Lz76 · 2023-07-05

**Soundness:** 3 good
**Presentation:** 3 good
**Contribution:** 2 fair
**Rating:** 5
**Confidence:** 4

**Summary:**

This paper studies visual OOD detection by introducing the textual outlier under the outlier exposure paradigm. Different from previous research focused on utilizing visual outliers, this work explores the benefits of textual outliers in the image domain. Specifically, they propose different ways to generate textual outliers based on the powerful GPT model. Comprehensive experiments have been conducted to demonstrate the effectiveness of textual outliers in OOD detection.

**Strengths:**

1. From the perspective of originality, to the best knowledge, this paper is the first work to explore the potential of textual exposure with multimodal neural networks for outlier exposure.
2. The overall presentation of this work is good, having intuitive illustrations and clear organization. The framework and the proposed textual outliers generation are easy to understand.
3. Instead of straightforwardly utilizing the LLM under the multimodal paradigm for visual OOD detection, this paper investigates the practical adaptation of different textual outlier types in the outlier exposure framework.
4. Experiments from different perspectives have been conducted to demonstrate the effectiveness of description-level textual outliers.

**Weaknesses:**

1. Regarding the technical level, exploring the textual information (like description level or caption level) under the multimodal setting for the image domain has been studied in [1]. The general paradigm, e.g., using LLM to generate the corresponding description or caption and then utilizing them in the image task, is similar.
2. Compared with the conventional outlier exposure, the optimizing objective and test time OOD detection have limited advanced design beyond the previous method.
3. Compared with the generation part, the filtering part seems to be more important as it has a more close relationship with the data quality for outlier exposure. However, the current filtering process is a little bit heuristic.
4. Considering the performance, the improvement compared with other advanced methods of using pure image outliers is not very significant (like Table 5 compared with DOE). Could the authors provide more explanation or discussion?

I generally appreciate the idea of introducing and excavating the potential of textual outliers. I hope the previous comments or the later questions can help the current contribution to be clearer and enhanced.

[1] Sachit Menon and Carl Vondrick. Visual classification via description from large language models. In The Eleventh International Conference on Learning Representations, 2023.


**Questions:**

Regarding the current version of the draft, I have the following question:
1. Could the authors discuss more the unique contribution of OOD detection beyond the general paradigm of utilizing textual information?
2. Could the authors provide more examples or system comparisons of the three different types of textual outlier generation?
3. Could the authors discuss more why the performance of textual OE is not consistently performing better than visual OE?

**Limitations:**

This paper has discussed the limitation, e.g., the proposed textual outlier method involves a heuristic selection process to refine the outputs of the generative model and also provided the potential solution in future work.

---

> ### Author Rebuttal · Authors · 2023-08-08
>
> Thank you for providing constructive feedback and suggestions. In particular, we appreciate your recognition of the novelty and efficiency of our work and the insightfulness of our analyses. Here, we hope that a more detailed explanation can address your concerns.
>
> > Q1 (W1, W2). Could the authors discuss more the unique contribution of OOD detection beyond the general paradigm of utilizing textual information?
>
> Our contribution is primarily centered around the proposition of textual outliers. Through a comprehensive analysis, we illuminate the potential of textual outliers as valuable indicators for detecting visual OoD detection (as elaborated in Section 3). We propose a range of textual outlier categories (outlined in Section 4) and identify key attributes of effective textual outliers (discussed in Section 5). As the reviewer pointed out, the distinctiveness of our method does not lie in the process of generating text through the LLM. Our focus is on utilizing these texts as textual outliers for the OoD detection task. Identifying informative textual outliers within the generated text demands a nuanced and intricate approach (W1). Our textual outlier approach is not constrained to using only uniform loss or energy scores, as demonstrated in our paper; it can also be applied with other objective functions or OoD scores. This flexibility enhances the scalability of our method (W2).
>
> Going beyond the utilization of textual outliers, our method exhibits computational efficiency by achieving competitive performance solely through training of a linear classifier. This advantage in computational efficiency can come in particularly handy when new OoD instances emerge frequently, and outlier exposure must be performed repeatedly.
>
> > W3. The current filtering process is a little bit heuristic.
>
> While our filtering approach includes heuristic elements, we address this by conducting ablation studies for each method to determine the optimal parameters (Appendix B.4, B.5).
>
> > Q2. Could the authors provide more examples or system comparisons of the three different types of textual outlier generation?
>
> Here are a few examples of textual outliers for ImageNet. We offer three distinct types of textual outliers for each class.
>
> Class: Warplane
> * word: “This is a photo of army”, “This is a photo of airport”, “This is a photo of airliner”
> * desc: “a photo of large and powerful", "a photo of designed for carrying weapons and other military equipment", "a photo of typically has a camouflage paint job",
> * caption: “a small plane with a window on the side”, “a silver airplane”, “a plane with a large window on the front”, “a plane taking off”
>
> Class: Greater Swiss Mountain Dog
> * word: “This is a photo of green”, “This is a photo of puppy”, “This is a photo of pets”
> * desc: "a photo of large, fluffy white dog", "a photo of black or brown markings on the face, ears, and tail", "a photo of long, thick coat"
> * caption: “two dogs are playing with each other”, “a dog with a white face”, “a dog with its tongue out”
>
> Based on the provided examples, our word-level outliers convey more abstract concepts. Similarly, caption-level outliers mostly contain descriptions of background elements or lack class-specific attributes. Description-level outliers include class-relevant information, but when the class label is omitted, they become very vague and difficult to interpret. In the revised manuscript, we will incorporate additional examples to enhance understanding.
>
> > Q3 (W4).  Could the authors discuss more why the performance of textual OE is not consistently performing better than visual OE?
>
> The results of using pure image outliers are presented in Table 5 under the label 'OE,' where our textual outlier approach notably outperforms the OE baseline. It should be noted that while DOE is not strictly categorized as a pure image outlier method. Therefore, the fact that textual outliers do not consistently outperform DOE does not imply that textual outliers do not consistently outperform pure image outliers. While DOE utilizes model perturbations, which are computationally demanding, our approach only requires training a lightweight linear classifier, making it computationally efficient. A key advantage of textual outlier exposure over DOE, albeit insufficiently highlighted in the current manuscript, lies in its computational efficiency. We will elaborate on this discussion in the revised manuscript to further underscore the benefits of adopting textual outlier exposure.

---

> > ### Comment · Reviewer_Lz76 · 2023-08-18
> > **Thanks for the response!**
> >
> > Thanks for the detailed response! Most of my concerns are well addressed by the further clarification and I appreciate the idea of exploring textual outliers in OOD detection.

---

### Official Review · Reviewer_T6K7 · 2023-07-06

**Soundness:** 3 good
**Presentation:** 3 good
**Contribution:** 3 good
**Rating:** 6
**Confidence:** 4

**Summary:**


This paper addresses the challenge of detecting Out-of-Distribution (OOD) data by introducing "textual outlier exposure" as an alternative to visual outliers. Instead of relying on visual examples, the authors explore the benefits of using textual equivalents in OOD detection. They propose various methods for generating textual outliers, which are validated through extensive experiments on large-scale OOD benchmarks. The findings demonstrate that the generated textual outliers outperform visual outliers and establish criteria for effective textual outliers, including their proximity to the data distribution, descriptiveness, and incorporation of visual semantics.

The contributions of this work include investigating the potential of textual outlier exposure with multi-modal neural networks, utilizing large-language models for generating textual outliers at different levels of detail, and validating their effectiveness in various OOD detection scenarios. The paper presents a novel and promising approach to OOD detection, showcasing the advantages of textual outliers over visual counterparts and providing valuable insights for designing impactful textual outliers.

**Strengths:**

- The paper presents an innovative and compelling idea, offering valuable insights for open-world learning, Out-of-Distribution (OOD) detection, and online learning. The findings of this study have significant implications for various research studies in these domains.

- The clarity and coherence of the paper are commendable, making it easy to comprehend and follow the authors' methodological approach.

- The inclusion of illustrative examples effectively enhances the understanding of key concepts.

- The figures and plots presented in the paper are visually clear, aiding in the visualization and interpretation of the experimental results.

- The paper offers comprehensive experimental studies, demonstrating the efficacy of the proposed method. The promising performance observed in these experiments further strengthens the validity and potential impact of the proposed approach.




**Weaknesses:**

While the paper presents detailed experimental studies, there are two notable weaknesses that should be addressed:

- The focus of the experimental studies is primarily limited to outlier exposure methods, neglecting the exploration of other types of Out-of-Distribution (OOD) detection methods. It would be valuable to consider and compare the proposed approach against alternative techniques in order to provide a more comprehensive evaluation.

- The omission of an ablation study to assess the impact of network architecture on the proposed methodology is a notable oversight. Investigating the influence of different network architectures on the performance of the proposed approach would enhance the understanding of its strengths and limitations.

**Questions:**


The experimental evaluation in your paper is commendable. However, I have two inquiries regarding the experiments:

- It appears that the recent state-of-the-art methods, such as ASH [1] and GradNorm [2], have not been included in your experimental comparisons. Could you provide insights into the rationale behind this omission? It would be valuable to report and compare the performance of your proposed method with these state-of-the-art approaches to provide a comprehensive assessment.

- Additionally, considering the impact of different network architectures on your method is crucial. Could you elaborate on how varying network architectures affect the performance and effectiveness of your proposed approach? Investigating this aspect would provide deeper insights into the robustness and generalizability of your method.

[1] Djurisic, Andrija, et al. "Extremely simple activation shaping for out-of-distribution detection." arXiv preprint arXiv:2209.09858 (2022).

[2] Huang, Rui, Andrew Geng, and Yixuan Li. "On the importance of gradients for detecting distributional shifts in the wild." Advances in Neural Information Processing Systems 34 (2021): 677-689.









**Limitations:**

No negative social impact.

---

> ### Author Rebuttal · Authors · 2023-08-08
>
> Thank you for acknowledging the novelty of our method and the extensive experiments. Here, based on your comments, we show various newly-conducted experimental results to further verify our method’s effectiveness.
>
> > Q1 (W1). It appears that the recent state-of-the-art methods, such as ASH [1] and GradNorm [2], have not been included in your experimental comparisons.
>
> Following the suggestion of the reviewer, we have compared our method with ASH and GradNorm, demonstrating superior performance in comparison to both. The table below presents the results of OoD detection performance, evaluated using AUROC, for a model trained on ImageNet1K across four OoD datasets. We would like to note that all the results are derived from the ResNet-101 architecture (BiT-S-R101 for GradNorm and ASH, CLIP-RN101 for our approach). In this experiment, we solely present results based on caption-level outliers for our method. In the revised manuscript, we will incorporate these results to offer a more comprehensive set of experiments.
>
> | | **iNaturalist** | **SUN** | **Places** | **Textures** | **AVG** |
> |----------|:---------------:|:-------:|:----------:|:------------:|:-------:|
> | GradNorm | 90.32 | 89.02 | 84.82 | 81.07 | 86.30 |
> | ASH | 94.84 | 88.72 | 86.61 | 88.40 | 89.64 |
> | ours | **95.54** | **91.84** | **88.42** | **91.70** | **91.93** |
>
> Additionally, for your reference, we have conducted and reported a comparative analysis of our method with post-hoc OoD detection methods in Appendix A.3, presented in Table 4.
>
> > Q2 (W2). Could you elaborate on how varying network architectures affect the performance and effectiveness of your proposed approach?
>
> Following the suggestion of the reviewer, we conducted experiments to compare the performance of our method across two different image encoder architectures, ResNet and ViT, the two architectures offered by CLIP. Importantly, our method consistently produces encouraging outcomes, even when applied to CLIP models built upon the ResNet architecture. We selected RN50x4, designed with a parameter size akin to that of ViT-B/32 (174M vs 176M). The performance analysis reveals a comparable trend between RN50x4 and ViT-B/32, resulting in AUROC scores of 86.43 and 87.55, respectively. For this experiment, we utilized ImageNet1K as the in-distribution dataset. The results are obtained from caption-level textual outliers.
>
> | | **parameter size** | **iNaturalist** | **SUN** | **Places** | **Textures** | **AVG** |
> |----------|:-------:|:---------------:|:-------:|:----------:|:------------:|:-------:|
> | RN50x4 |174M| 91.69 | 85.19 | 82.93 | 85.94 | 86.43 |
> | ViT-B/32 |176M| 94.55 | 86.59 | 81.51 | 79.48 | 87.55 |
>
> In the revised manuscript, we will incorporate these results to offer a more comprehensive set of experiments.

---

> > ### Comment · Reviewer_T6K7 · 2023-08-17
> >
> > Thanks the authors for providing responses to my questions and concerns.
> > Would you please provide the FPR values of your reported experiments as it is a very important metric in OOD method performance?

---

> > > ### Author Response · Authors · 2023-08-18
> > >
> > > Thank you for expressing interest in further results. Presented below are tables summarizing the FPR and AUROC values for two experiments, each involving a comparison with recent state-of-the-art methods and a different architecture.
> > >
> > > > for Q1
> > >
> > > |     | **iNaturalist** |   | **SUN** |   | **Places** |  | **Textures** |  | **AVG** |   |
> > > |----------|:-----------------:|:-------:|:---------:|:-------:|:------------:|:-------:|:--------------:|:-------:|:---------:|:-------:|
> > > |          | FPR95             | AUROC | FPR95     | AUROC | FPR95        | AUROC | FPR95          | AUROC | FPR95     | AUROC |
> > > | GradNorm | 50.03           | 90.32 | 46.48   | 89.02 | 60.86      | 84.82 | 61.41        | 81.07 | 54.69   | 86.30 |
> > > | ASH      | 30.95           | 94.84 | 56.17   | 88.72 | 59.17      | 86.61 | 52.75        | 88.40 | 49.76   | 89.64 |
> > > | ours     | **20.19**           | **95.54** | **46.17**   | **91.84** | **52.23**      | **88.42** | **50.62**        | **91.70** | **42.30**   |**91.93** |
> > >
> > > > for Q2
> > >
> > > | | **iNaturalist** |  | **SUN** | | **Places** |  | **Textures** |  | **AVG** |   |
> > > |:--------:|:---------------:|:--------:|:-------:|:--------:|:----------:|:--------:|:------------:|:--------:|:-------:|:-------:|
> > > |          | FPR95             | AUROC    | FPR95     | AUROC    | FPR95        | AUROC    | FPR95          | AUROC    | FPR95     | AUROC   |
> > > | RN50x4   | 51.53           | 91.69    | 64.13   | 85.19    | 66.29      | 82.93    | 57.93        | 85.94    | 59.97   | 86.43 |
> > > | ViT-B/32 | 32.92           | 94.55    | 55.68   | 86.59    | 70.54      | 81.51    | 74.29        | 79.48    | 58.35 | 87.55   |
> > >
> > > Indeed, the FPR results exhibit a similar trend as the AUROC results.

---

> > > > ### Comment · Reviewer_T6K7 · 2023-08-20
> > > >
> > > > Thanks the authors for providing responses to my questions and concerns. I have no more questions now and would go forward to discussion with other reviewers and change my score accordingly.

---

### Author Rebuttal · Authors · 2023-08-08


We sincerely appreciate the reviewers' time and invaluable feedback. The unanimous consensus among reviewers highlights our paper's insightful contribution on textual outliers (T6K7, Lz76, JSZe, yebH). The reviewers also commend the novelty and significance of the addressed problem (T6K7, JSZe, yebH), along with acknowledging the method's effectiveness through extensive experiments (T6K7, Lz76, JSZe). Additionally, we are pleased that the reviewers found the paper clear and easily understandable (Lz76, JSZe, yebH).

We respond to each reviewer's comments in detail below. We will incorporate the reviewers' suggestions into the manuscript revisions, which we believe will significantly enhance the paper's strength.

---

### Decision · Program_Chairs · 2023-09-21

**Decision:**

Accept (poster)

**Comment:**

Congratulations to the authors on their work. The reviewers unanimously vote for acceptance. Please add the new results into the final version of the paper.